# Dental Profile of Brazilian Patients with Rare Skeletal Genetic Disorders: Clinical Features and Associated Factors

**DOI:** 10.3390/healthcare12101046

**Published:** 2024-05-19

**Authors:** Ivanete Cláudia de Oliveira Vilar, Jennifer Reis-Oliveira, Gabriela Lopes Ângelo Dornas, Mauro Henrique Nogueira Guimarães de Abreu, Natália Cristina Ruy Carneiro, Ana Cristina Borges-Oliveira

**Affiliations:** 1Department of Community and Preventive Dentistry, School of Dentistry, Universidade Federal de Minas Gerais, Belo Horizonte 31270-901, Brazil; claudiavilar2014@gmail.com (I.C.d.O.V.); jenniferreisdeoliveira@hotmail.com (J.R.-O.); gabriela.angelo@hotmail.com (G.L.Â.D.); anacboliveira7@gmail.com (A.C.B.-O.); 2Department of Pediatric Dentistry, School of Dentistry, Universidade Federal de Minas Gerais, Belo Horizonte 31270-901, Brazil; nataliacrcarneiro@hotmail.com

**Keywords:** dental care, dental care for people with disabilities, dental health, people with disabilities

## Abstract

The aim of this study is to compare the dental profiles of Brazilian patients with rare genetic skeletal disorders and normotypical patients. A cross-sectional study was carried out with 210 individuals aged between 2 and 54 years old [105 with rare diseases (Mucopolysaccharidosis/MPS n = 27 and Osteogenesis Imperfecta/OI n = 78) and 105 without rare diseases] and their parents/caregivers. The parents/caregivers answered a questionnaire about individual aspects of their child and the dental profile was identified from questions related to dental history and the presence/absence of dental problems. The patients’ oral cavity was also examined by three examiners for dental caries, malocclusion, gingivitis, and dental anomalies. The average age of individuals with a rare disease was 14.1 years (±12.2) and the median was 9.5 years. Participants who had already used the public health system (SUS) dental care services had a 2.24 times higher chance of belonging to the group with a rare disease (OR = 2.24; 95% CI: 1.07–4.89). Patients with rare diseases are 14.86 times more likely to have difficulty receiving dental treatment (OR = 14.86; 95% CI: 5.96–27.03) and 10.38 times more likely to have one or more dental problems (OR = 10.38; 95% CI: 1.95–35.17). Individuals with rare disorders have a greater history of difficulty in accessing dental treatment, using the SUS, and were diagnosed with more dental problems compared to normotypical individuals.

## 1. Introduction

According to the World Health Organization (WHO), rare diseases are defined as all diseases that have a prevalence of less than 65 cases per 100,000 inhabitants [1]; the majority of these diseases having a genetic origin. It is estimated that there are six to eight thousand types of rare diseases, which can affect different organs; each with their own symptoms and therefore requiring specific treatments [2,3]. In Brazil, there are about 13 million people diagnosed with a rare disease [4].

Mucopolysaccharidoses (MPS) and osteogenesis imperfecta (OI) are two types of rare genetic disorders that affect skeletal development. MPS is caused by a deficiency in specific enzymes involved in the degradation of glycosaminoglycans (GAGs). The progressive accumulation of GAGs in the body triggers an inflammatory response causing damage to cells and tissues, compromising the functions of organs and systems [5,6,7,8]. OI is caused by an alteration in the production of type 1 collagen, with a prevalence of 1:10,000 [9]. Clinical manifestations of OI comprise early deafness, blue sclera, and brittle bones [10,11,12].

In addition to systemic health problems, individuals diagnosed with MPS or OI may also present with several alterations in the oral cavity. Previous studies have shown that malocclusion, facial alterations, and dental anomalies are common alterations found in these populations. Additionally, dentinogenesis imperfecta (DI) is a dental anomaly highly prevalent in OI individuals [11,13,14,15,16,17,18,19,20,21]. These two groups of rare diseases are also considered more vulnerable to developing dental caries and to experiencing 0poor oral hygiene [22]. 

Regarding health care assistance, in 2014 the Brazilian government published the Policy of Comprehensive Care for People with Rare Diseases within the Unified Health System (*Sistema Único de Saúde* [SUS]) [23]. The SUS is the public health system in Brazil and aims to achieve universal access to healthcare services for the population. The Policy of Comprehensive Care for People with Rare Diseases aims to ensure better access conditions for patients with rare diseases to health services, highlighting the role of primary health care (PHC) in patient care and as the main entry for these users to the SUS [24]. Depending on the severity of the rare disease and the limitations of the patient, dental care can be provided via PHC for preventive procedures and less complex interventions. However, the existence of a public health policy aimed exclusively at patients with rare diseases does not guarantee that access to dental services is fully contemplated. Recent studies carried out in Brazil have shown that people with rare diseases have less access to oral health services compared to people without diseases [20,25]. 

Thus, there is a need to better understand the oral health conditions and the dental profile of individuals with rare genetic diseases in order to reflect on and ensure a better quality of access to the public services offered. 

In this context, the aim of the present study was to compare the dental profile of Brazilian patients with rare genetic skeletal disorders and normotypical patients. We hypothesize that individuals with rare genetic diseases have less access to oral health services and have a higher prevalence of oral problems when compared to normotypical individuals.

## 2. Materials and Methods

### 2.1. Ethical Aspects

This article is in accordance with the Strengthening the Reporting of Observational Studies in Epidemiology (STROBE) statement [26]. This study received approval from the Human Research Ethics Committee of the Federal University of Minas Gerais (certificate numbers: 01480212.4.0000.5149 [MPS] and 54755516.4.0000.5149 [OI]).

### 2.2. Study Design, Setting, and Sample

A cross-sectional study was carried out with a sample of 210 individuals (105 with rare diseases and 105 without rare diseases) between 2 and 54 years of age and their parents/caregivers. A convenience sample was selected of individuals with two rare genetic diseases affecting skeletal development: MPS and OI, and a sample without rare genetic diseases, paired through sex and age (1:1). In order to expand the number of participants, the snowball sampling recruitment technique was employed [27]. The study was conducted between January and December 2019.

The group with rare diseases was recruited from five Brazilian states (Ceará, Espírito Santo, Minas Gerais, Rio de Janeiro, and São Paulo). The volunteers receive medical assistance at hospitals in each state that are reference centers for the treatment of these two diseases. Individuals without rare diseases were recruited from outpatient clinics at these hospitals. The hospitals belong to the Brazilian Unified Health System (SUS).

### 2.3. Eligibility Criteria

The following inclusion criteria were considered for participants:Individuals diagnosed with MPS or OI and their parents/guardians;Individuals without rare diseases and without other clinical/sensory diagnosis

(physical/intellectual disability, syndromes, autism, chronic/acute illnesses, neurodegenerative injuries) and parents/guardians;

3.Individuals two years of age or older and their parents/guardians.

The following exclusion criteria were followed:4.Individuals with OI or MPS and individuals without rare diseases who refused to undergo the clinical dental examination, and their parents/guardians;5.Individuals with OI or MPS and individuals without rare diseases whose parents/guardians refused to answer the questionnaire.

### 2.4. Data Collection

Data was carried out through a structured questionnaire about the sociodemographic and behavioral aspects of individuals with and without rare diseases. Dental history was assessed through the respective questions on: the patient’s dental experience, the presence of dental pain (last 12 months), when their last dental visit was, the reason for the child’s last visit to the dentist, the use of SUS dental care services, their satisfaction with the dental care assistance at the last visit, any difficulties experienced in receiving dental treatment for the child [28]. The questionnaire was answered by parents/caregivers. A clinical oral examination was also performed and assessment of the presence/absence of oral problems was noted.

Dental caries was assessed according to the WHO diagnostic criteria [29]. The number of decayed (presence of cavitated lesion), missing, and filled teeth was recorded for primary and permanent teeth. The presence/absence of malocclusion was identified according to the following criteria: the presence of overjet (increased/protrusion, anterior crossbite, absent), overbite (increased/deep bite, anterior open bite, absent, top), and posterior crossbite [25,29,30]. The presence of gingivitis was recorded based on the analysis of the contour and color of the gingiva [31]. The presence of dental anomalies was recorded (conical teeth, agenesis, microdontia, tooth rotation, and others). Dental agenesis was identified as a possible diagnosis because intraoral radiographs were not performed in the present study. Developmental defects of enamel and DI were also assessed [25,29,30,32].

### 2.5. Calibration Process

Prior to the main study, a calibration process was performed. It was divided into theoretical and practical phases and three examiners were trained by a dental specialist in the area, considered a gold standard. 

The theoretical phase was carried out through images of malocclusions, dental anomalies, gingivitis, and dental caries. This was conducted to verify inter- and intra-examiner diagnostic agreement, with an interval of seven days between the two training moments. Based on the kappa values obtained (0.74 to 1.00), it was verified that the examiners were trained to carry out the practical stage of the calibration.

Practical calibration was performed in one of the hospitals selected for the main study (Belo Horizonte, Minas Gerais, Brazil). Five individuals with MPS/OI and five individuals without rare diseases participated in this phase. From the results, the kappa index was calculated. Values ranged from 0.78 to 1.00 (malocclusion = 1.00, dental anomaly = 0.97, dental caries = 0.96, and gingivitis = 0.78). The kappa test results were very good, and the examiners were considered trained to start the main study. 

### 2.6. Pilot Study

The pilot study was carried out after the calibration process, and five individuals with rare diseases, five without rare diseases, and their respective parents/caregivers participated in this phase. Data collection was performed at the previously selected public hospitals. The results indicated that changes in the methodological procedures were deemed unnecessary. Consequently, participants in the pilot study were included in the final study sample.

### 2.7. Statistical Analysis

Data analysis was carried out using the software Statistical Package for the Social Sciences (SPSS for Windows, version 26.0, SPSS Inc., Chicago, IL, USA). Univariate, bivariate (chi-square, *p* < 0,05), and multivariate analyzes (logistic regression) were performed. The dependent variable was the genetic condition (With rare disease/Without rare disease). The independent variables were household income, patient’s dental experience, dental pain (last 12 months), last dental appointment, reason for the child’s last visit to the dentist, use of SUS dental care services, satisfaction with the care received at the last dental appointment, difficulties for the child to receive dental treatment, presence/absence of dental problems. Independent variables were included into the multivariate model in accordance with their statistical significance (*p* < 0.25, backward stepwise selection).

## 3. Results

The age of the participants ranged from 2 to 54 years, with an average of 14.1 years (±12.2) and a median age of 9.5 years. Among the individuals with rare diseases, 78 (37.1%) were with OI and 27 (12.9%) with MPS. Most of the sample were from the State of Minas Gerais, Brazil, [98(46.7%)], followed by the states of Espírito Santo [56(26.7)], São Paulo [22(10.5)], Rio de Janeiro [20(9.5)], and Ceará [14(6.7%)].

Table 1 summarizes the statistical association between the presence/absence of rare disease and the patient’s characteristics. The variables with associations were the presence of dental pain (last 12 months), when the last dental appointment took place, the use of SUS dental care services, satisfaction with dental care assistance at the last visit, difficulties for the child to receive dental treatment, and the presence/absence of dental problems (*p* < 0.05).

Table 2 summarizes, according to parents’/caregivers’ reports, the reasons why their children faced difficulties in receiving dental care.

Multiple logistic regression analysis revealed that participants who had already used SUS dental care services had a 2.24 times higher chance of belonging to the group with rare diseases (OR = 2.24; 95% CI: 1.07–4.89). Patients with a history of difficulties receiving dental treatment were 14.86 times more likely to belong to the group with rare diseases (OR = 14.86; 95% CI: 5.96–27.03). Patients diagnosed with one or more dental problems were 10.38 times more likely to belong to the group with rare diseases (OR = 10.38; 95% CI: 1.95–35.17) (Table 3).

## 4. Discussion

Individuals with rare diseases may face difficulties obtaining sufficient oral health assistance in both public and private systems. Some of the factors described in the literature as being involved in patients facing those difficulties are the health professionals’ lack of knowledge about the rare disease, difficulties in obtaining transportation, the health professional’s fear of intervening [33], the lack of acceptance, the parents’ lack of knowledge about the importance of their child receiving oral health care, as well as some specific oral characteristics of the condition [25,30].

The hypothesis of the present study has been confirmed: individuals with rare diseases experience more difficulties receiving dental treatment and more dental problems when compared to the normotypical group. In the present study, it was possible to observe that the majority of the participants with rare diseases face difficulties in receiving dental treatment (87.9%). This data points to the existence of a problem that can enhance the worsening of the oral conditions of these patients and implying a more complex dental attendance [34]. In the present study, it was also possible to show, through parents’ and caregivers’ reports, that the main factors related to the lack of access to dental services were the difficulties in obtaining transportation to take the child to the dental office and finding a dentist who understands the problem of their child. This highlights the existence of barriers that hinder access to oral health assistance for individuals with rare diseases.

Our results corroborate previous studies published by Iriart et al. [33] and Debossan et al. [25]. The first of these [33] was a Brazilian multicenter qualitative study that aimed to analyze the ‘therapeutic itineraries’ of patients in search of diagnosis and treatment for rare genetic diseases in two cities. Twenty-eight interviews were conducted with patients and caregivers and among the reports we can highlight the lack of knowledge of non-genetic doctors about rare diseases and the difficulties in obtaining transportation to access health specialists. In the second one [25], a cross-sectional study also developed in Brazil, the authors evaluated access to oral health care services for individuals with rare genetic diseases. They showed that individuals with rare genetic diseases have less access to oral healthcare. The authors also described that the lack of health professionals willing to provide the necessary care and the lack of family information about the importance of oral health care for individuals with rare diseases can be considered important factors involved in the lack of access to dental services.

Additionally, the present study also showed that individuals with rare diseases presented with a higher prevalence of dental problems compared to individuals without rare diseases. Previous studies had already stated that individuals with rare genetic diseases such as MPS and OI present more oral alterations such as malocclusion, dental caries, and dental anomalies [35,36,37]. Most of these alterations are related to the clinical consequence of the disease. However, it is also important to highlight that the lack of dental assistance targeted to these populations, the lack of information for caregivers about the importance to accessing oral healthcare, and the need for better-qualified professionals to offer dental services, may also contribute to the appearance of new oral alterations or the worsening of a pre-existing condition [22,38]. Thus, it is evident that the importance of acting more effectively in these determining factors can enable an improvement in the conditions of access to healthcare and, as a consequence, enhance the quality of life of these patients.

Another important result arrived at in the present study was that the majority of participants with rare diseases use the Brazilian public health system, SUS, to access oral healthcare. In Brazil, the main entrance for patients with rare diseases to receive dental care is the same as that used by all other users, that is, through the PHC which constitutes a part of the SUS [39]. Primary care plays an important role in improving the quality of life of patients with MPS and OI. It takes place through the Family Health Strategy (ESF) by carrying out an active search to establish the user’s link with the rare condition and their family members and ensures the performance of the PHC in offering continuous care. The oral health team should be able to welcome this patient and listen to them in a humane way and carry out health promotion through preventive measures such as meticulous guidance on oral hygiene, supervised brushing, carrying out prophylaxis, applying fluoride, and identifying caries in its early stages. In addition, if it is not possible to provide care to the user at the basic health unit, the team should be responsible for referring them to specialized care; this being of great importance after care in secondary or tertiary units, along with the patient’s counter-referral to the PHC to perform an adequate follow-up.

Despite the existence of a public health system and policies for the inclusion and expansion of healthcare for people with rare diseases in Brazil [23], these individuals still face difficulties and barriers to access to oral health services. It is important to emphasize that these difficulties can be mitigated by putting into practice the health policies already provided for in Brazilian law and ensuring that they are actually being implemented in practice.

Some limitations of the present study should be acknowledged. Dental profiles were assessed through a questionnaire answered by parents/caregivers, and this implies a risk of recall bias. Also, this was conducted as a cross-sectional study, which makes it difficult to infer any causal relationship between the presence of rare disease and the results found. A convenience sample was chosen, and sample size estimation was not calculated. This limitation was due to a low overall prevalence of MPS and OI making large sample recruitment quite challenging. However, the authors used the snowball sampling technique in an effort to expand the number of participants as much as possible. On top of that, the authors were concerned to include a comparison group without rare genetic diseases in order to reduce the possible influence of combined characteristics on the association between independent variables.

The results of the present study play an important role for the scientific community and the MPS and OI communities. With the obtained results, knowledge about the difficulties of Brazilian patients with rare diseases in receiving dental care is improved. This knowledge may result in decision-making and generate effective actions on expanding access to oral health services, as well as contributing to the improvement of the quality of life of these patients.

## 5. Conclusions

Individuals with rare diseases have a greater history of difficulty in accessing dental treatment, use the public health system/SUS, and are diagnosed with more dental problems compared to normotypical Brazilian patients. The results shown here suggest the need to put into practice oral health services for people with the rare conditions as already provided for by public health law.

## Figures and Tables

**Table 1 healthcare-12-01046-t001:** Association between the presence/absence of rare disease, family income, and the dental profile of Brazilian patients (n = 210).

Individual Variables		Rare Disease			*p* Value *
		Present n (%)	Absent n (%)	Total n (100%)	
Family income	≤1 minimum wage	18 (48.6)	19 (51.4)	37	0.856
	>1 minimum wage	87 (50.3)	86 (49.7)	173	
Patient’s dental experience	Yes	95 (49.0)	99 (51.0)	194	0.298
	No	10 (62.5)	6 (37.5)	16	
Dental pain (<12 months)	Presence	40 (66.7)	20 (33.3)	60	**0.002**
	Absence	65 (43.3)	85 (56.7)	150	
Last dental visit **	≥2 years	18 (72.0)	7 (28.0)	25	**0.014**
	<2 years	77 (45.6)	92 (54.4)	169	
Reason for the child’s last visit to the dentist **	Preventive dental care	66 (50.8)	64 (49.2)	130	0.475
	Dental treatment (restorative, periodontal, endodontic, surgical)	29 (45.3)	35 (54.7)	64	
Use public dental care service **	Yes	73 (59.8)	49 (40.2)	122	**<0.001**
	No	22 (30.6)	50 (69.4)	72	
Satisfaction with the last dental care visit **	Yes	15 (71.4)	6 (28.6)	21	**0.029**
	No	80 (46.2)	93 (53.8)	173	
Difficulties for the child to receive dental treatment **	Yes	58 (87.9)	8 (12.1)	66	**<0.001**
	No	37 (28.9)	91 (71.1)	128	
Dental problems	Presence	102 (57.0)	77 (43.0)	179	**<0.001**
	Absence	3 (9.7)	28 (90.3)	31	

* X^2^ Test (significance level of 5%/value in bold): statistical significance <0.05). ** 16 participants have never had a dental appointment.

**Table 2 healthcare-12-01046-t002:** Difficulties in receiving dental treatment as reported by parents/caregivers (n = 194 *).

	Rare Disease		
What Are the Reasons that Made It Difficult for Your Child to Receive Dental Treatment?	Present n (%)	Absent n (%)	Total n (100%)
There were no difficulties	37 (28.9)	91 (71.1)	128
It is difficult to find a dentist who understands my child’s condition	20 (95.2)	1 (4.8)	21
It’s hard to pay for my son’s dental treatment	2 (66.7)	1 (33.3)	3
It’s hard to find someone to take care of my other children	4 (66.7)	2 (33.3)	6
My child is afraid of the dentist	3 (100)	0	3
Difficulty getting transportation to take my son	14 (77.8)	4 (22.2)	18
My child does not cooperate during dental treatment + It is difficult to find a dentist who understands my child’s condition + It’s hard to pay for my son’s dental treatment	1 (100)	0	1
My child does not cooperate during dental treatment + It is difficult to find a dentist who understands my child’s condition + It’s hard to find someone to take care of my other children + Difficulty getting transportation to take my son	1 (100)	0	1
It is difficult to find a dentist who understands my child’s condition + Difficulty getting transportation to take my son	8 (100)	0	8
It is difficult to find a dentist who understands my child’s condition + It’s hard to find someone to take care of my other children	5 (100)	0	5
TOTAL	95	99	194

* 16 participants have never had a dental appointment.

**Table 3 healthcare-12-01046-t003:** Multivariate model of logistical regression explaining the dental profile between patients with rare diseases.

Independent Variables	Unadjusted OR (95% CI)	Adjusted OR (95% CI)	*p* Value *
Use of SUS dental care services.			0.041
No	1.0	1.0	
Yes	3.38 (1.82–6.28)	2.24 (1.07–4.89)	
Difficulties for the child to receive dental treatment			<0.001
No	1.0	1.0	
Yes	17.83 (7.75–40.98)	14.86 (5.96–27.03)	
Presence/absence of dental problems			0.006
Absence	1.0	1.0	
Presence	12.36 (3.62–42.16)	10.38 (1.95–35.17)	

OR: odds ratio; 95% CI: confidence interval. * statistical significance (<0.05).

## Data Availability

The data that support the findings of this study are available on request from the corresponding author. The data are not publicly available due to restrictions, e.g., because they contain information that compromises the privacy of research participants. All listed authors meet the authorship criteria, and all authors agree with the content of the manuscript.

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
