# Peer review of "Dental Profile of Brazilian Patients with Rare Skeletal Genetic Disorders: Clinical Features and Associated Factors"

_healthcare, 2024, doi:10.3390/healthcare12101046_

Round 1

Reviewer 1 Report

Comments and Suggestions for Authors

Dear Author,

This is the review of the article "Dental profile of Brazilian patients with skeletal rare genetic 2 disorders: clinical features and associated factors"

Abstract:

- Mention something about the clinical examination (No. Examiner, evaluation system)

- I recommend placing keywords in alphabetical order

Aim: I recommend moving the working hypothesis below, after line 70, for better clarity.

The introduction was well formulated, supported by references.

Material and method:

- According to what formula did you choose the number of 210 subjects from the group with genetic disease?

- Criteria of inclusion and exclusion?

- Was the questionnaire used validated? Does it have a name?

- In line 117, add "Brasil" after the name of the hospital and the city

- The rest of the information is very well presented

The results are well presented.

The discussions are well expressed, with correct references.

Conclusions: well formulated, supported by results.

Author Response

1) Abstract: Mention something about the clinical examination (No. Examiner, evaluation system)

Authors´response: The authors would like to thank the reviewer for the careful revision of the manuscript. We added the information about clinical examination at line 21 of the abstract.

“Oral cavity was also examined by three examiners for dental caries, malocclusion, gingivitis, and dental anomalies.”

2) I recommend placing keywords in alphabetical order

Authors´response: Thank you for your comment. We made the corrections at line 27.

“Keywords: dental care; dental care for disabled; dental health; disabled persons.”

3) Aim: I recommend moving the working hypothesis below, after line 70, for better clarity.

Authors´response: Thank you for your comment. We move the hypothesis to line 75 from the introduction section.

“In this context, the aim of the present study was to compare the dental profile of Brazilian patients with skeletal rare genetic disorders and normotypical patients. We hypothesize that individuals with rare genetic diseases have less access to oral health services and have a higher prevalence of oral problems when compared to normotypical individuals.”

4) Material and method: According to what formula did you choose the number of 210 subjects from the group with genetic disease?

Authors’ response: In the present study we did not use formula to choose the number of subjects to be included in the study. We choose a convenience sample as considering the rarity of the diseases investigated and their low overall prevalence, which make the use of sampling estimations quite challenging. Participants were recruited from the outpatient medical hospital that are reference centers in rare diseases in Brazil. Moreover, in order to expand the number of participants, we used the snowball sampling recruitment technique. This information was added in Material and Methods section, line 90.

“In order to expand the number of participants, the snowball sampling recruitment technique was employed [27]. The study was conducted between January and December 2019.”

5) Criteria of inclusion and exclusion?

2.3 . Eligibility criteria

Authors´response: We added the information about eligibility criteria ate line 97 from material and methods section.

“The following inclusion criteria were considered for participants:

1- Individuals diagnosed with MPS or OI and their parents/guardians.

3- Individuals without rare diseases and without other clinical/sensory diagnosis

(physical/intellectual disability, syndromes, autism, chronic/acute illnesses, neurodegenerative injuries) and parents/guardians.

4- Individuals two years of age or older and their parents/guardians.

The following exclusion criteria were followed:

1- Individuals with OI or MPS and individuals without rare diseases who refused to under-go through the clinical dental examination, and their parents/guardians.

2- Individuals with OI or MPS and individuals without rare diseases whose parents/guardians refused to answer the questionnaire.”

6) Was the questionnaire used validated? Does it have a name?

Author’s response: Thank for your comment. We used a structured questionnaire that was developed from a broad study of the literature on the topic and based on previous research25,28 to collect information about sociodemographic characteristics, behavioral aspects, and dental history. To ensure the internal reliability of the instrument, a test-retest was carried out at an interval of seven days.

7) In line 117, add "Brasil" after the name of the hospital and the city

Authors´response: We added the word “Brazil” as the reviewer suggested at line 117 from Materials and Methods section.

“Practical calibration was performed in one of the hospitals selected for the main study (Belo Horizonte, Minas Gerais, Brazil).”

Reviewer 2 Report

Comments and Suggestions for Authors

The purpose of the study was to compare the dental profile of Brazilian patients with skeletal rare genetic disorders and normotypical patients.

Please the authors of the following corrections:

1. Abstarct - methodology and results are repeated, how many individual examinees and the age of the examinees, please leave the same only in one place (with 105 individuals with rare genetic disorders mucopolysaccharidosis [MPS (n=27)] and osteogenesis imperfecta [OI (n=78) )], with 2-54 years old and their parents/caregiver OR being 78 (37.1%) with OI and 27 (12.9%) with MPS.)

2. Abstract - results: list the differences in caries, malocclusions, gingivitis and other important factors between the examined groups, but how many are there and their ages?! Based on that, write a conclusion.

3. Methodology - if you stated how you follow the STROBE guidelines - where was the study conducted, in what period, sample size, inclusion and exclusion criteria...?

4. Results - where is the p value in tbl 3?

5. Discussion - describe in more detail the limitations of the study and its strength?

Author Response

1) Abstract - methodology and results are repeated, how many individual examinees and the age of the examinees, please leave the same only in one place (with 105 individuals with rare genetic disorders mucopolysaccharidosis [MPS (n=27)] and osteogenesis imperfecta [OI (n=78) )], with 2-54 years old and their parents/caregiver OR being 78 (37.1%) with OI and 27 (12.9%) with MPS.)

Authors´response: The authors would like to thank the reviewer for the careful revision of the paper. We made the corrections at abstract section and excluded the repeated information.

2) Abstract - results: list the differences in caries, malocclusions, gingivitis and other important factors between the examined groups, but how many are there and their ages?! Based on that, write a conclusion.

Authors’ response: The authors would like to thank the reviewer for the comment. However, according to the journals guidelines, the limit of words that must contain in the Abstract section is 200. After other corrections suggested by the reviewers, we achieved 199 words. We believe that information essential for understanding the study may be lost if other sections of the abstract are removed.

3) Methodology - if you stated how you follow the STROBE guidelines - where was the study conducted, in what period, sample size, inclusion and exclusion criteria...?

Authors´response: The authors would like to thank the reviewer for the comment. We added the proper information following the STROBE guidelines:

-As previously stated at line 91, the study was conducted at five Brazilian states (Ceará, Espírito Santo, Minas Gerais, Rio de Janeiro and São Paulo).

- Information about the period of the study was addressed at line 90 from materials and methods section: “The study was conducted between January and December 2019”.

- Samples size: In the present we choose a convenience sample as considering the rarity of the diseases investigated and their low overall prevalence, which make the use of sampling estimations quite challenging. Participants were recruited from the outpatient medical hospital that are reference centers in rare diseases in Brazil. Moreover, in order to expand the number of participants, we used the snowball sampling recruitment technique. This information was added in Material and Methods section, line 90.

 “A convenience sample was selected of individuals with two rare genetic diseases affecting skeletal development: MPS and OI, and a sample without rare genetic diseases, paired through sex and age (1:1). In order to expand the number of participants, the snowball sampling recruitment technique was employed [27].”

- We added the information about eligibility criteria ate line 97 from material and methods section.

“The following inclusion criteria were considered for participants:

1- Individuals diagnosed with MPS or OI and their parents/guardians.

3- Individuals without rare diseases and without other clinical/sensory diagnosis

(physical/intellectual disability, syndromes, autism, chronic/acute illnesses, neurodegenerative injuries) and parents/guardians.

4- Individuals two years of age or older and their parents/guardians.

The following exclusion criteria were followed:

1- Individuals with OI or MPS and individuals without rare diseases who refused to under-go through the clinical dental examination, and their parents/guardians.

2- Individuals with OI or MPS and individuals without rare diseases whose parents/guardians refused to answer the questionnaire.”

4) Results - where is the p value in tbl 3?

Authors´response: We added the information about p value in table 3, line 194-195.

“Use SUS dental care services: p=0.041; Difficulties for the child to receive dental

Treatment: p<0.001; Presence/absence of dental problems: p=0.006.”

6) Discussion - describe in more detail the limitations of the study and its strength?

Authors’ response: The authors would like to thank the reviewer for the comment. We provided more information about the sample recruitment in the limitations and the efforts to minimize possible bias, line 268.

“A convenience sample was chosen, and sample size estimation was not calculated. This limitation was due to a low overall prevalence of MPS and OI making large sample recruitment quite challenging. However, the authors used the snowball sampling technique in an effort to expand the number of participants as much as possible.”

Round 2

Reviewer 2 Report

Comments and Suggestions for Authors

As far as I'm concerned, it can be accepted for publication.